# Comparison of X-Ray Imaging and Computed Tomography Scan in the Evaluation of Knee Trauma

**DOI:** 10.3390/medicina55100623

**Published:** 2019-09-23

**Authors:** Mustafa Avci, Nalan Kozaci

**Affiliations:** Department of Emergency Medicine; University of Health Sciences, Antalya Education and Research Hospital, Antalya 07100, Turkey; nalankozaci@gmail.com

**Keywords:** computed tomography, emergency department, fracture characteristics, knee injury, X-ray

## Abstract

*Background and objectives*: The aim of the study was to compare the accuracy of X-ray (XR) imaging according to computed tomography (CT) scanning in the diagnosis of knee bone fractures, and in the determination of fracture characteristics, and to identify CT scan indications in patients with knee trauma. *Materials and methods*: The patients who presented to the emergency department (ED) due to knee trauma between January 2017 and December 2018 and who underwent XR imaging and CT scans were included in the study. XR images were reinterpreted by an emergency physician. The official reports, which had been interpreted by a radiologist in the hospital automation system for CT images, were considered valid. *Results*: Five hundred and forty-eight patients were included in the study. Of the patients, 200 (36.5%) had fractures in XR imaging and 208 (38.0%) had fractures in CT scans. Compared to CT scanning, XR imaging was found to have 89% sensitivity, 95% specificity, 92% positive predictive value, and 92% negative predictive value in identifying the fracture. The sensitivity of XR imaging in identifying growth plate fracture, angulation, stepping off, and extension of the fracture into the joint space was determined as 78% and less. According to the kappa value, there was determined a perfect concordance between the XR imaging and CT scans in angulation, stepping off, and extension of the fracture into the joint space. This concordance was moderate in growth plate fractures. *Conclusions*: XR imaging has a low sensitivity in identifying knee fractures. There is a moderate concordance between XR imaging and CT scanning in identifying growth plate fractures. Therefore, CT scanning should be performed in patients whose fracture type and fracture characteristics are not able to be determined exactly with XR imaging in knee injury.

## 1. Introduction

Acute knee injury is a common presentation to the emergency department (ED), which comes about mostly due to sports activities, traffic accidents, and falls from height. Delayed diagnoses of overlooked knee fractures may cause limitation of movement, knee instability, angular deformity, or persistent pain. Therefore, early and accurate diagnosis is important [1,2,3].

X-ray (XR) is the first and most used imaging modality to evaluate knee bone trauma [3,4,5]. In decisions regarding XR imaging requirements for knee trauma, Ottawa knee rules and Pittsburgh knee rules are used to reduce radiation exposure [2,6,7]. In daily practice, anterior–posterior and lateral XR imaging is performed in at least two planes to evaluate the knee trauma [2,3,6]. Internal oblique, external oblique, tunnel, or sunrise (skyline, axial, or tangential) XRs can be performed if required [6]. Unfortunately, obtaining additional images in patients with severe injuries can be difficult, or even impossible. Nowadays, computed tomography (CT) scanning is used as a common imaging method, following XR imaging, to evaluate the bone trauma [3]. However, due to it having an expensive cost and a high radiation dose, CT scanning is recommended in selected patients only [8,9].

There are studies comparing the imaging methods in the literature. Some of these studies are about comparing the imaging methods, which are: XR and magnetic resonance imaging (MRI) [1,10], XR imaging and ultrasonography (USG) [11,12,13,14,15,16], USG and CT scan [17], CT scan and MRI [18], USG and MRI [19], and some are about XR imaging and CT scans [3,5,8,9,20,21,22]. A small number of the studies comparing XR imaging and CT scanning examine the knee joint. Current studies have focused on the presence of fractures, and the localization of the fractured bone. In addition, there are no studies conducted in the ED comparing knee joint imaging methods.

The aim of this study was to compare the accuracy of XR imaging according to CT scans in the diagnosis of knee bone fractures, localization of fractured bone, determination of fracture type (fissure, linear, spiral, fragmented, etc.), determination of fracture characteristics (extension of the fracture into the joint space, growth plate fractures, angulation, and stepping off),and to identify CT scan indications in patients presented to the ED due to knee trauma.

## 2. Materials and Method

The study was approved by the University of Health Sciences Antalya Education and Research Hospital Clinical Research Ethics Committee (Antalya, Turkey) with the registry number 10/8 on 28March 2019. Following the approval of the ethics committee, the patients who presented to the ED of a tertiary hospital due to knee trauma between January 2017 and December 2018, and who underwent XR imaging and a CT scan, were included in this retrospectively designed study. Patients’ information, XR, and CT images were obtained from the hospital automation system and Pacs database. Patients of all ages, including children, were included in the study. Patients who had fixators (internal or external) in knee bones, or who had knee joint arthroplasty, or whose XR imaging or CT scan could not be obtained from the Pacs database, were excluded from the study.

A standard data record form was created for the study. The demographic data of the patients, reinterpretations of the XR and CT images, treatment methods (operation, reduction, or splint) applied to the patients, and the outcome of the patients (hospitalization or discharge from ED) were recorded on this form.

The XR images were reinterpreted by an emergency medicine specialist who had nine years of experience. All of the XR images were reinterpreted by the same emergency medicine specialist and the eight-step modified Kozaci protocol was used to standardize the XR imaging reinterpretations (Table 1) [11]. The emergency-medicine specialist who reinterpreted the XR images was blind to CT interpretation. The CT interpretations, which had been interpreted by a radiologist following the CT scan, were evaluated according to the modified Kozaci protocol, and recorded.

In the hospital where the study was conducted, extremity CT scanning is routinely performed using the HITACHI-ECLOS multislice 16 ch (Japan) device, with sagittal, coronal, and three-dimensional reformatted images. The device used for routine XR imaging is the USX-RAY brand device that has the conventional recent technology U-arm HF radiographic system.

### Statistical Analysis

Twenty-three patients who had knee joint arthroplasty, 16 patients who had fixators (internal or external) in knee bones, and 9 patients whose XR imaging or CT scan could not be obtained from the Pacs database, were excluded from the study. Analysis of the data collected in the study was performed using the Statistical Package for the Social Sciences 22 statistical software package (IBM Corporation, IL, USA). The sensitivity, specificity, positive predictive value (PPV), negative predictive value (NPV), and Kappa (κ) coefficient of the XR imaging were calculated and compared to the CT scans. Concordance was graded according to the κ coefficient. A κ value of greater than 0.75 was considered as perfect concordance, 0.75 to 0.40 was considered as a moderate concordance, and less than 0.40 was considered as a poor concordance [23]. To determine the statistical significance and assumptions of the predictions, *P* < 0.05 with 95% confidence intervals was considered significant in all analyses. For descriptive statistics, the data obtained using chi-square test and κ statistics were compared.

## 3. Results

Five hundred and forty-eight patients were included in the study. Of the patients, 202 (36.9%) were female and 346 (63.1%) were male. The mean age of the patients was 43.17 ± 19.67 years. Of the patients, 200 (36.5%) had fractures in XR imaging, and 208 (38.0%) had fractures in CT scans. The most commonly fractured bone was the tibia (Table 2). Forty-three percent of the tibia fractures were tibial plateau fractures. Compared to the CT scans, XR imaging was found to have 89% sensitivity, 95% specificity, 92% PPV, and 92% NPV (area under curve (AUC): 0.919, confidence interval (CI): 0.890–0.947) in identifying the fracture (Figure 1).

In identifying the fracture, the lowest sensitivity of XR was found in femoral fractures (67%), and the highest was found in patella fractures (100%) (Table 2).

In CT scans, 28 (5.1%) patients had concomitant adjacent bone fractures (Table 3) (Figure 2). Of these patients, 24 (4.4%) had two bone fractures, and 4 (0.7%) had three bone fractures. XR imaging could not identify any of three simultaneous bone fractures. The sensitivity and specificity of XR imaging in identifying two or three simultaneous bone fractures were found to be 50% and 100%, respectively (AUC: 0.748, 95% Cl: 0.628–0.8768). According to the κ value, there was determined a moderate concordance between the XR imaging and the CT scans in identifying concomitant adjacent bone fractures.

In the CT scans, the most common fracture type was determined as a fragmented fracture (21.5%) (Figure 2). Of the patients who were determined as having a fragmented fracture in a CT scan, in XR imaging, 68 (58%) were interpreted as having a fragmented fracture, 18 (15%) were interpreted as a linear fracture, 16 (14%) were interpreted as a fissure-type fracture, 10 (8%) were interpreted as a spiral fracture, 2 (1.6%) were interpreted as an avulsion-type fracture, and 4 (3.4%) were interpreted as having no fracture. The highest sensitivity of XR imaging in determining a fracture type was calculated as the avulsion-type fracture. The sensitivity of XR imaging was determined to be low in linear-, fragmented-, spiral-, and fissure-type fractures. According to the κ value, there was determined a perfect concordance between the XR imaging and the CT scan in avulsion-type fractures. This concordance was moderate in linear and fragmented fractures, and was poor in fissure and spiral fractures (Table 4).

The sensitivity of XR imaging in identifying growth plate fracture, angulation, stepping off, and extension of the fracture into the joint space was determined as 78% or less. According to the κ value, there was determined a perfect concordance between the XR imaging and the CT scan in angulation, stepping off, and extension of the fracture into the joint space. This concordance was moderate for growth plate fractures (Table 5).

Sixty-three (11.5%) patients were younger than 18 years old. Eight (12.7%) of these patients were determined to have growth plate fractures in the CT scan. All of these fractures were in the tibia. The sensitivity of XR imaging in identifying the growth plate fracture was determined as 75%.

Of the patients, 156 (28.5%) were hospitalized, and 392 (71.5%) were discharged from the ED.

## 4. Discussion

There may be cases where XR is insufficient to evaluate bone fractures in knee injuries. In particular, a CT scan may be required for the treatment plan. In the studies performed in acute trauma, the sensitivity of XR imaging in identifying bone pathologies according to CT scanning was found to be between 75–83%. While the sensitivity of XR imaging is high in superficial bones such as the patella and the distal femur, this rate is very low in tibial plateau fractures [2,5,18,20]. In our study, tibia fractures were identified most commonly, and 43% of the tibia fractures were tibial plateau fractures. The overall sensitivity, PPV, and NPV of XR imaging compared to CT scanning in identifying the knee bone fractures for all bones was 89%, 92%, and 92%, respectively. The highest sensitivity of XR imaging was identified in patella fractures, as 100%. In contrast, the sensitivity of XR imaging was low in tibia (sensitivity: 80%) and femoral fractures (sensitivity: 67%). In our study, the reason for higher overall sensitivity can be attributed to the distance between the bones of the knee joint (distal femur, proximal tibia, proximal fibula, and patella bones), which is more distant than the bones of the ankle joint (distal tibia, distal fibula, and talus bones). As the distance between the bones forming the joints decreases, the overlapping structures may make it difficult to interpret the images. In addition, since the tibial plateau is located deeply in the knee joint, investigating the isolated tibial plateau fracture may be more difficult than examining the four bones that make up the knee joint.

Concomitant adjacent bone fracture is important in bone fractures, because it may change the treatment decision. In a study conducted in knee trauma, 20% of the patients had two simultaneous bone fractures. Fifty-six of these two simultaneous bone fractures were identified in the tibial lateral condyle and proximal fibula, five were identified in the distal femur and tibial plateau, and 13 were identified in the proximal fibula with fibular avulsion [3]. In a study conducted on ankle trauma, concomitant adjacent bone fracture was identified in 63 (20%) patients. In this study, the sensitivity of XR imaging in identifying two simultaneous bone fractures was reported as 56%, while the specificity was reported as 94%; the sensitivity in identifying trimalleolar fractures was reported as 17%, and the specificity was reported as 100% [5]. In another study investigating 205 patients with long bone traumas, nine patients had radius and ulna, and one patient had tibia and fibula; in addition to the long bone fracture, two patients had patella fractures, one patient had clavicula fractures, and one patient had calceneus fractures [12]. In a similar study investigating metatarsal fractures in 72 patients, multiple metatarsal fractures were detected in four patients [13].In a study performed on foot and ankle trauma, three patients had concomitant adjacent bone fractures [16]. In another study on metacarpal trauma, concomitant adjacent bone fracture was seen in 10% of the patients [14]. In our study, 4.4% of the patients had two simultaneous bone fractures. In addition, distal femur, proximal fibula, and proximal tibia fractures were present simultaneously in four patients. The sensitivity of XR imaging was calculated to be 50% in identifying two or three simultaneous bone fractures, and XR imaging could not identify any of three simultaneous bone fractures. It was determined that the sensitivity of XR imaging decreased gradually as the number of fractured bones increased. This may be attributed to the increased deterioration of the anatomy with the increase in the number of fractured bones. As deterioration of the anatomy increases, the interpretation of XR imaging becomes more difficult, and the error rate becomes high.

Treatment decisions are made according to the characteristics of fractures in patients who have fractures due to knee trauma. The extension of the fracture into the joint space, angulation, stepping off, presence of concomitant adjacent bone fractures, and fracture type (fissure, linear, fragmented, spiral, avulsion) are evaluated in XR imaging. CT scanning is performed when the characteristics of the fracture cannot be identified in XR imaging. For example, in proximal tibial fractures, the fractures extending to the joint space longer than 2 mm, open fractures, and the medial condyle fractures should be fixed under surgery. In contrast, the lateral condyle fractures and the fractures not extending to the joint space can be treated non-operatively [24]. In a study investigating fracture characteristics, the highest sensitivity of XR imaging in determining the fracture type was found in linear fractures (55%). In the same study, the sensitivity of XR imaging was calculated as very low in fragmented-, spiral-, avulsion-, and fissure-type fractures [5]. In our study, the most common fracture type was the fragmented fracture (21.5%) in CT scans. Only 58% of the fragmented fractures identified in CT scans were interpreted as fragmented fractures in XR imaging. The rest were interpreted as a different type of fracture, or as no fracture. The highest sensitivity of XR imaging in determining the fracture type was found in the avulsion-type fractures (69%). However, the sensitivity was calculated to be less than 60% in linear-, fragmented-, spiral-, and fissure-type fractures. According to the κ value, there was determined a perfect concordance between the XR imaging and CT scans in the avulsion-type fractures. This concordance was moderate in linear and fragmented fractures, and was poor in other fractures. These results indicate that XR imaging is insufficient, and may be a cause of misinterpretations in determining the fracture type.

Angulation and the distance of stepping off in fractures are important in both reduction and operation decisions. In addition, the extension of the fracture into the joint space and the presence of the growth plate fracture can change the treatment decision. In one study, compared to a CT scan, the sensitivity of XR imaging in determining the angulation and the stepping off was found to be very low—as low as 56% and 49%, respectively [5]. In a study comparing XR imaging to CT scans in investigating the extension of the long bone fractures into the joint space, it was concluded that diaphyseal fractures of the distal femur and all metaphyseal fractures should be imaged in CT scans, to visualize the extension of the fracture into the joint space [21]. In a similar study comparing XR imaging to CT scanning, the sensitivity of XR imaging in determining the extension of the fracture into the joint space was found to be 48% [5]. In a study comparing XR imaging to CT scanning in growth plate fractures of the distal tibia, it was found that the sensitivity of XR imaging was lower than 90% in fractures involving the metaphysis, while it was 64% in the separation of the articular surface, 61% in the dorsal formation of the articular surface, and 79% in subluxation. In the same study, while there was no misclassification identified in Salter–Harris type I and II fractures, the highest misclassification was in Salter–Harris type III fractures [22].In a similar study comparing XR imaging to CT scanning, 44 (14%) patients were found to be younger than 18 years old, and 13 (30%) of these patients had growth plate fractures in CT scans. In the same study, the sensitivity of XR imaging in determining the growth plate fracture was identified as very low—as low as 54% [5]. In our study, the sensitivity of XR imaging in determining angulation, stepping off, and the extension of the fracture into the joint space was identified as 75%, 71%, and 78%, respectively. In addition, according to the κ value, there was identified a perfect concordance between XR imaging and CT scanning in determining angulation, stepping off, and the extension of the fracture into the joint space, which was 0.811, 0.774, and 0.782, respectively. Furthermore, 11.5% of the patients in our study were younger than 18 years old. A growth plate fracture was determined in a CT scan in 12.7% of these patients. All of the growth plate fractures were in the proximal tibia. The sensitivity of XR imaging compared to CT scanning in determining a growth plate fracture was identified as 75%, and according to the κ value, there was a moderate concordance of 0.746 between the imaging methods.

Trauma may cause soft-tissue injuries in addition to bone fractures. Post-traumatic hemarthrosis in knee joints suggest an important intra-articular injury. The most common cause of hemarthrosis in knee joints is anterior cruciate ligament (ACL) rupture, either alone or in combination with other injuries [10]. Magnetic resonance imaging is used for the evaluation of soft tissues, but this is not usually possible in ED. Ultrasonography is used to visualize bone fractures after acute trauma in the ED [11,17], and to diagnose soft tissue injuries, ACL rupture with hematoma, and hemarthrosis [16,19,25]. Therefore, the use of ultrasonography as a preferred imaging method in the ED is becoming widespread.

## 5. Conclusions

XR imaging has a low sensitivity in identifying knee bone fractures. It is also insufficient to determine the type of fracture, and may lead to misinterpretations. The sensitivity of XR imaging is much lower in the case of two or three simultaneous fractures. In addition, according to the kappa value, there is a moderate concordance between XR imaging and CT scanning in identifying growth plate fractures. Therefore, CT scanning should be performed in patients whose fracture type and fracture characteristics are not able to be determined exactly with XR imaging in knee injuries. However, more extensive studies are needed.

## 6. Limitation

XR and CT images were reinterpreted being unaware of the clinical findings and bone mineral density of the patients. These might have caused misinterpretation in both imaging methods. In addition, the number of patients that were 18 years old and younger was relatively low in our study; with larger studies, better results can be obtained about growth plate fracture.

## Figures and Tables

**Figure 1 medicina-55-00623-f001:**
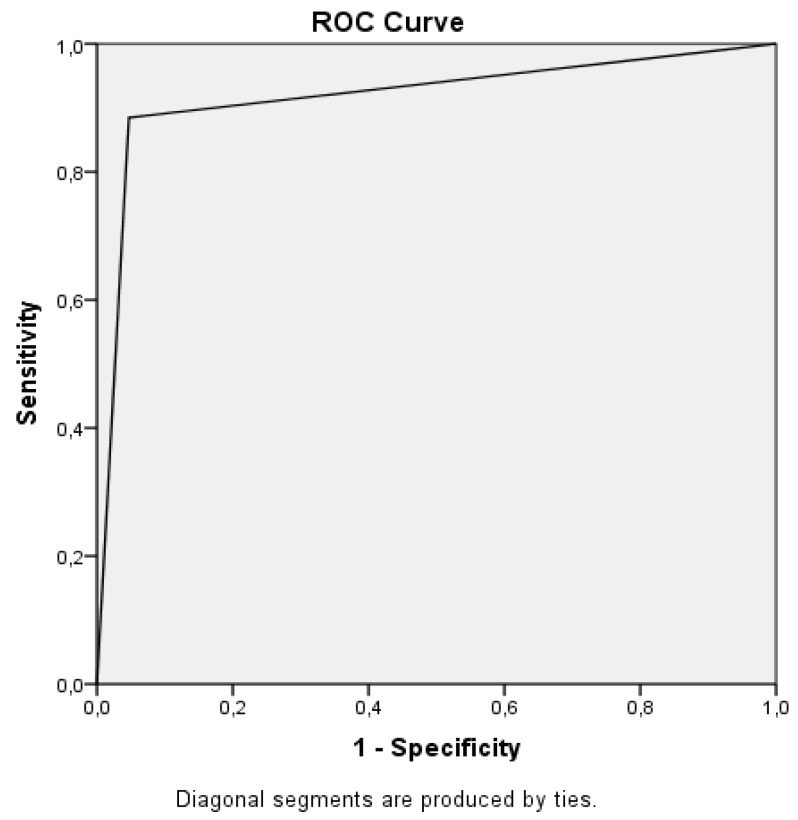
Receiver Operating Characteristic (ROC) curve showing the sensitivity and specificity ratio of X-ray imaging in identifying the knee fractures.

**Figure 2 medicina-55-00623-f002:**
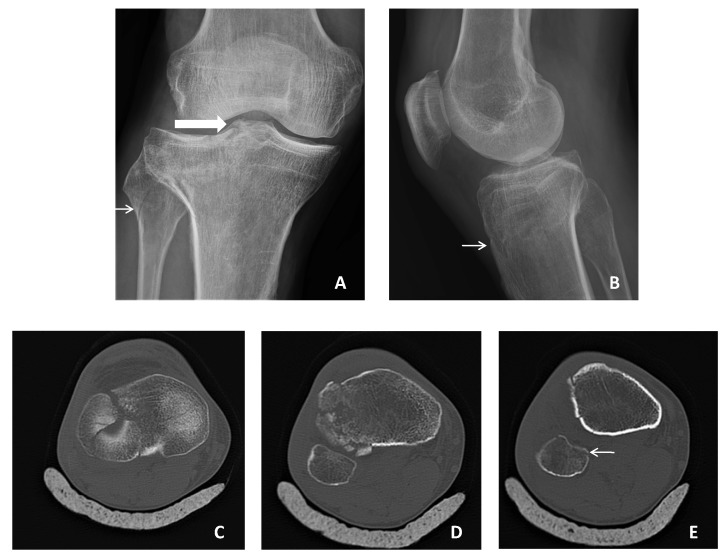
The knee images of a 58-year-old male patient. (**A**) Anterior–posterior XR image, thin arrow: fissure type fracture in fibula, thick arrow: tibia plateau fracture; (**B**) lateral XR image, arrow: a compression in tibia (linear fracture); (**C**) axial CT image, a fragmented fracture in tibia that extends into the joint space; (**D**) axial CT image, the fragmented fracture in tibia; (**E**) axial CT image, arrow: a linear fracture in fibula.

**Table 1 medicina-55-00623-t001:** The interpretation protocol of X-ray images (modified Kozaci protocol) [10].

1	Detection presence of fracture (cortical deterioration)
2	Determine the type (fissure, linear, fragmented, torus) and localization of fracture.
3	Measure the degree of angulation of the fracture.
4	Measure the distance of stepping off.
5	Is there an extension of the fracture into the joint space or epiphyseal line?
6	Does the fracture include the epiphyseal line? (Growth plate fracture?)
7	Detect the presence of concomitant adjacent bone fracture.
8	Control of the joint space and the presence of joint dislocation.

**Table 2 medicina-55-00623-t002:** The knee fractures which were identified with X-ray imaging and computed tomography scan, and the sensitivity and specificity of X-ray imaging in identifying the fractured bone.

Bone	XR, N (%)	CT, N (%)	Sensitivity/Specificity	AUC (95% CI)
**Femur**	28 (5.1)	42 (7.7)	67/100	0.833 (0.746–0.921)
**Tibia**	120 (21.9)	130 (23.7)	80/96	0.881 (0.839–0.923)
**Patella**	48 (8.8)	46 (8.4)	100/100	0.998 (0.995–1.000)
**Fibula**	18 (3.3)	22 (4.1)	82/100	0.909 (0.813–1.000)

XR: X-ray; CT: computed tomography; AUC: area under the curve; CI: confidence interval.

**Table 3 medicina-55-00623-t003:** Concomitant adjacent bone fractures.

Bone	XR, N (%)	CT, N (%)
**Femur + patella**	2 (0.4)	4 (0.7)
**Femur + tibia**	-	4 (0.7)
**Femur + fibula**	-	2 (0.4)
**Tibia + patella**	2 (0.4)	2 (0.4)
**Tibia + fibula**	12 (2.2)	12 (2.2)
**Femur + fibula + tibia**	-	4 (0.7)

XR: X-ray; CT: computed tomography.

**Table 4 medicina-55-00623-t004:** Diagnostic accuracy of XR imaging in determining the fracture type.

Type of Fracture	XR, N (%)	CT, N (%)	Sensitivity/Specificity	AUC (95% CI)	Kappavalue
Fissure	48 (8.8)	22(4.0)	55/93	0.739 (0.609–0.868)	0.305
Linear	40 (7.3)	24 (4.4)	58/95	0.767 (0.644–0.889)	0.405
Spiral	24 (4.4)	18(3.3)	44/97	0.707 (0.556–0.858)	0.357
Fragmented	68 (12.4)	118 (21.5)	58/100	0.788 (0.731–0.845)	0.681
Avulsion	20 (3.6)	26 (4.7)	69/100	0.844 (0.736–0.952)	0.773

XR: X-ray; CT: computed tomography; AUC: area under the curve; CI: confidence interval.

**Table 5 medicina-55-00623-t005:** Diagnostic accuracy of XR imaging in determining the fracture characteristics.

Fracture Characteristics	XR, N (%)	CT, N (%)	Sensitivity/Specificity	AUC (95% CI)	Kappa Values
Extension of the fracture into the joint space	150 (27.4)	176 (32.1)	78/96	0.872 (0.834–0.910)	0.782
Growth plate fracture	8 (1.5)	8 (1.5)	75/100	0.873 (0.693–1.000)	0.746
Angulation	90 (16.4)	118 (21.5)	75/100	0.871 (0.823–0.918)	0.811
Stepping off	94 (17.2)	126 (23.0)	71/99	0.852 (0.804–0.901)	0.774

XR: X-ray; CT: computed tomography; AUC: area under the curve; CI: confidence interval.

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
