# Peer review of "Comparison of X-Ray Imaging and Computed Tomography Scan in the Evaluation of Knee Trauma"

_medicina, 2019, doi:10.3390/medicina55100623_

Round 1

Reviewer 1 Report

I appreciate modifications and earlier reponses. The manuscript has been improved however following points need more clarifications.
point3: "A κ value of greater than 0.75 was considered as perfect concordance, 0.75 to 0.40 as moderate concordance, and less than 0.40 as poor concordance. [11]"
11. Kartal ZA, Kozaci N, Çekiç B, Beydilli İ, Akçimen M, Güven DS, et al, Computed tomography interpretations in multiply injured patients: comparison of emergency physicians and on-call radiologists, Am J Emerg Med (2016), http://dx.doi.org/10.1016/j.ajem.2016.08.044
Apparently you are refereeing to your own 2016 paper for this criteria which is not acceptable. You may refer to a statistical book that has mentioned such thresholds [0.75, 0.4].

Point5: "In our study, the reason of higher overall sensitivity can be attributed to the distance between the bones of the knee joint to each other is more distant than the bones of the ankle joint. As the distance of the bones forming the joints decreases, the overlapping structures may make it difficult to interpret the images."
The sentence is not very clear. Please identify the name of the bones you mean that are close to each other in the ankle but far in the kneed joint.

Point6: Could you please describe briefly the novelty of your study in your response letter. This should be couple of sentences. Readers should know why you have repeated a similar study. For example PMCID: PMC3139878 has done similar work in tibial plateau.

Reviewer 2 Report

The revision improved the quality of the manuscript, however, there is still need for a major revision. The revised sentences were not marked with different color which made the identification of the revised parts very difficult! Especially the page and line numbers were not correct: Page 9, Line 245-246- there is no such line on Page 9.

I have concerns how could the results be biased that the X-ray was reported by an emergency physician and not by a radiologist (, too). It is unclear what the authors mean: "The official reports which had been interpreted by a radiologist in the hospital automation system for CT images were considered valid.". Why should it be invalid? Please clarify. Introduction still does not provide much novel data, especially the 2nd paragraph which clearly demonstrates that "CT scan has higher sensitivity and specificity than XR imaging in identifying of the bone fracture" which makes the relevance of the aims in the next paragraph questionable: "The aim of this study was to compare the accuracy of XR imaging according to CT scan in the diagnosis of knee bone fracture". In the new Fig2, image A, the tibia plateau fracture should be also marked with an arrow, which is clearly seen also on XR, that it is extending into the joint space.  My previous concern was not answered correctly. Patients of all ages were included in the study. Please refer if including children could not
bias the data?

Round 2

Reviewer 2 Report

I accept the revised version of the manuscript.

This manuscript is a resubmission of an earlier submission. The following is a list of the peer review reports and author responses from that submission.

Round 1

Reviewer 1 Report

The authors of this manuscript compared the fracture evaluations of over 400 knee trauma patients using XR imaging and CT scan. The main aim was to study sensitivity and specificity level of XR imaging technique in such fractures. The manuscript is written well. Before further steps toward publication, following points need to be considered in the revised version:

Materials and methods, page 3, line 81. “A κ value of greater than 0.75 was considered as 81 perfect concordance, 0.75 to 0.40 as moderate concordance, and less than 0.40 as poor concordance”. Please add a reference for this criteria. A k value around 0.75 does not look like perfect.

Results, page 5, line 115. Is there any particular reason that kappa value was not calculated for fracture types in Table3? Please also clarify this point in the methods.

Discussion, page 6, line 145. Please add your hypothesis about the difference between your overall calculated XR sensitivity and the values calculated in reference 6 (89% vs. 75%).

Discussion. Please add few sentences to explain the novelties of your study. You mentioned that earlier studies exist that have investigated the XR imaging performance compared with CT in traumatic knee joints.

Discussion. Please add one paragraph to the discussion section about the limitations of the current study and the potential next steps.

Reviewer 2 Report

Albeit the paper is well written, iIt is already well known that CT scan has high sensitivity and specificity in identifying bone pathologies in acute trauma compared to XR especially in tibial plateau fractures. There is no mention on ultrasound, albeit it plays an important role in the evaluation of soft tissues. No radiological image is presented. 

The authors compared the XR and CT imaging in knee trauma and demonstrated that XR imaging has a low sensitivity in identifying knee fractures, especially in tibial  plateau fractures, which is not very novel data. The sensitivity of XR imaging is much lower in case of two or three simultaneous fractures. Authors conclude that CT scan should be  performed in patients who thought to have complex knee injury, which should be extended with novel data. Therefore, the paper should undergo a major revision highlighting the novel findings compared to the literature.

Specific comments:

1. Introduction is too general, including general, well-known statements which should be revised thoroughly.

2. English language should be revised (eg., XR imagings)

3. Patients of all ages were included in the study. Please refer if including children could not bias the data?

4. Could the BMD loss of the patients influence the fracture identification on XR compared to CT scan? 

5. Who was the reader (board certified radiologist?) of the XR?

6. Some examples should be demonstrated with XR and CT images.
In summary, the paper should be thoroughly revised in order to highlight its novel findings compared to the well-known facts.